# Mitochondrial sAC-cAMP-PKA Axis Modulates the ΔΨ_m_-Dependent Control Coefficients of the Respiratory Chain Complexes: Evidence of Respirasome Plasticity

**DOI:** 10.3390/ijms242015144

**Published:** 2023-10-13

**Authors:** Rosella Scrima, Olga Cela, Michela Rosiello, Ari Qadir Nabi, Claudia Piccoli, Giuseppe Capitanio, Francesco Antonio Tucci, Aldo Leone, Giovanni Quarato, Nazzareno Capitanio

**Affiliations:** 1Department of Clinical and Experimental Medicine, University of Foggia, 71122 Foggia, Italy; olga.cela@unifg.it (O.C.); michela.rosiello@unifg.it (M.R.); ari.nabi@su.edu.krd (A.Q.N.); claudia.piccoli@unifg.it (C.P.); aldoleone.1994@gmail.com (A.L.); 2Department of Biology, College of Science, Salahaddin University-Erbil, Erbil 44001, Kurdistan, Iraq; 3Department of Translational Biomedicine and Neuroscience, University of Bari “Aldo Moro”, 70124 Bari, Italy; giuseppe.capitanio@uniba.it; 4European Institute of Oncology, Istituto di Ricovero e Cura a Carattere Scientifico (IRCCS), 20141 Milan, Italy; francescoantonio.tucci@ieo.it; 5Treeline Biosciences, San Diego, CA 92121, USA; giovanni.quarato@gmail.com

**Keywords:** mitochondrial respiratory chain complexes, supercomplexes, cAMP/PKA signaling pathway, soluble adenylate cyclase, metabolic flux theory, mitochondrial membrane potential

## Abstract

The current view of the mitochondrial respiratory chain complexes I, III and IV foresees the occurrence of their assembly in supercomplexes, providing additional functional properties when compared with randomly colliding isolated complexes. According to the plasticity model, the two structural states of the respiratory chain may interconvert, influenced by the intracellular prevailing conditions. In previous studies, we suggested the mitochondrial membrane potential as a factor for controlling their dynamic balance. Here, we investigated if and how the cAMP/PKA-mediated signalling influences the aggregation state of the respiratory complexes. An analysis of the inhibitory titration profiles of the endogenous oxygen consumption rates in intact HepG2 cells with specific inhibitors of the respiratory complexes was performed to quantify, in the framework of the metabolic flux theory, the corresponding control coefficients. The attained results, pharmacologically inhibiting either PKA or sAC, indicated that the reversible phosphorylation of the respiratory chain complexes/supercomplexes influenced their assembly state in response to the membrane potential. This conclusion was supported by the scrutiny of the available structure of the CI/CIII_2_/CIV respirasome, enabling us to map several PKA-targeted serine residues exposed to the matrix side of the complexes I, III and IV at the contact interfaces of the three complexes.

## 1. Introduction

The mitochondrial respiratory chain is constituted by inner membrane-embedded respiratory complexes (RCs) (CI, NADH dehydrogenase; CII, succinate dehydrogenase; CIII, Cytochrome c reductase/bc1; and CIV, cytochrome c oxidase), sequentially transferring reducing equivalents to dioxygen, which is reduced to H_2_O [1]. Part of the available redox energy is harnessed by CI, CIII and CIV to pump protons from the matrix side to the intermembrane space, thereby establishing an electrochemical proton gradient largely constituted by the electrical component ΔΨ_m_ [2]. The proton motive force is, henceforth, utilized to drive endergonic reactions, mainly the ATP synthesis using the H^+^-F_o_F_1_-ATP synthase (CV).

The long-standing view of RCs as independent entities functionally connected by the freely diffusible CoQ and cytochrome c as mobile redox carriers (fluid model) [3] has been disputed over the last two decades, regarding the emerged evidence that the mitochondrial respiratory chain complexes interact with each other and form stable supercomplex (SC) assemblies with defined stoichiometric ratios (solid model) [4]. However, neither model satisfactorily accounts for the conflicting experimental evidence and the discrepancies are not yet solved [5,6]. The in-situ identification of the fraction of RCs organized in SCs is a challenging task, and the occurrence of artifacts in the ex-situ experimental procedures cannot be ruled out.

An alternative model foresees a balanced distribution between free RCs and SCs with the two pools in a dynamic equilibrium (plastic model) [7,8,9]. This opens the possibility that the balance between RCs and SCs might be controlled by physiological/metabolic cues [10].

An alternative to tackling the controversy is to use a functional approach. The flux control theory is a mathematical framework that models the contribution of the distinct enzymatic steps to the overall metabolic flux pursued by the pathway, wherein those steps are components [11,12,13]. Each step can be graded with a specific control coefficient ranging theoretically from zero to one if its strength in controlling the overall flux varies from a negligible to an absolute contribution, respectively. The control coefficient of a specific step can be assessed experimentally by comparing the activity of the overall flux with that of the isolated selected step, both titrated independently with the same concentrations of a specific inhibitor of the enzymatic step. The “summation corollary” of the theory proposes that the sum of the control coefficients of all the steps contributing to the pathway must closely approach unity [11,12]. If the sum of the control coefficients is higher than the unity, this means that two or more enzymatic steps will work in a functional supramolecular complex (Appendix A).

The identification of the enzymatic step(s) kinetically controlling the overall respiratory flux and, consequently, the oxidative phosphorylation (OxPhos) has been the object of extensive investigations within the framework of the “metabolic flux control theory” [14,15,16]. The ensued results, however, provided inhomogeneous outcomes. The reasons for the discrepancies proved to rely on: (i) the integrity of the biological sample; (ii) the choice of the respiratory substrate(s); and (iii) the energetic state of the mitochondrial membrane.

On this basis, in previous studies using intact cells, we reported that the sum of the control coefficients of the mitochondrial RCs significantly exceeded the unity value when the oxygen consumption rate was titrated with inhibitors of either CxI, CxII, CxIII or CxIV under uncoupled conditions (i.e., in the presence of FCCP or valinomycin) or conditions utilizing the membrane potential (ΔΨ_m_) (i.e., state III of respiration). Conversely, under conditions preserving the ΔΨ_m_ (i.e., state IV of respiration or in the presence of the CV inhibitor oligomycin), the sum of the CI–CIV control coefficients was below one [17,18]. Our interpretation was that the energy state of the mitochondrial inner membrane influenced the formation of the SCs from a pool of isolated complexes. In particular, the collapse of the ΔΨ_m_ promoted the assembly of CxI, CxIII and CxIV. This could be envisioned as an adaptive response of the OxPhos system to the cellular bioenergetics demand with the enhancement of the ΔΨ_m_ taken as a signal of low proton motive force-consuming F_o_F_1_-ATP-synthase activity. Under these conditions, supercomplexes would disassemble in their RC constituents, thereby slowing respiration and sparing reducing substrates. The opposite would occur if the proton motive force was utilized (with a substantial reduction in the ΔΨ_m_ for ATP synthesis). In this case, the formation of respirasomes was favoured with enhanced respiration to fulfil the greater energy needs.

The post-translational modification of proteins plays a well-recognized and amply documented important role for rapid and transient changes in the structure of protein-modulating enzyme activity as well as interfering or favouring protein–protein interactions. A recent extensive proteomic analysis has revealed a previously unexpected and wide range of post-translational modifications of the mitochondrial proteome, including all the components of the OxPhos machinery [19]. Reversible phosphorylation of practically all the RCs has been reported to be the most common covalent modification [20,21,22], consistent with the presence of kinases and phosphatases in the mitochondria for both serine/threonine and tyrosine residues [23,24]. However, a large body of evidence is available about the functional impact of phosphorylation/dephosphorylation of individual RCs (some time conflicting). To the best of our knowledge, only a few reports have directly or indirectly investigated the role of post-translational modifications in the assembly and function of SCs.

In the present study, we extended our previous observations and carried out a systematic analysis on the inhibitory titration curves to investigate the impact of the cAMP/PKA signalling cascade on the distribution of the flux control coefficients among the proton motive RCs and the influence of this on the membrane energy state.

## 2. Results

### 2.1. The Respiratory Flux Control Coefficient of CIV Is Influenced by the Mitochondrial Respiratory State and the Activity of PKA

Intact HepG2 cells were grown in the presence of calf serum and assayed for endogenous mitochondrial respiratory activity using high-resolution respirometry (Figure 1A,B). Two conditions were selected, namely resting and FCCP-uncoupled respiration. Compared to the resting conditions, the presence of the protonophore FCCP resulted in a two-fold increase in the oxygen consumption rate (OCR) and a complete collapse of the ΔΨ_m_ (see Figure 3A,B). The oxygen consumption rates under both conditions (OCR_Rest_ and OCR_Unc_) were titrated with increasing concentrations of the cytochrome c oxidase inhibitor KCN. As shown in Figure 1B, the normalized OCR_Rest_ and OCR_Unc_ resulted in different inhibitor titration profiles, which were particularly evident at the lower KCN concentrations, with the OCR_Rest_ being significantly more resistant to KCN inhibition compared to the OCR_Unc_. The difference can be better visualized in a plot correlating the percentages of the activities attained for the resting and uncoupled respiration at the same concentration of KCN (Figure 1C). The results confirmed what was already reported [17,18], but serve as an internal control for the other conditions tested throughout the present study.

To investigate if the PKA-mediated signalling pathway, which was likely activated by the continuous exposure of the HepG2 cells to the serum and had some effect on their respiratory activities, we carried out the above-described titration assay using HepG2 cells pre-treated with the PKA inhibitor H89 (Figure 1D). The results obtained and illustrated in Figure 1E,F showed that the H89 treatment abolished the differences in the KCN-titration profiles when the coupled and uncoupled OCRs were compared with the OCR_Rest_ losing its apparent higher resistance to KCN. Accordingly, the graph correlating the OCR_Rest_ and OCR_Unc_ resulted in data points that were situated on the diagram diagonal (Figure 1F). It is important to note that when the HepG2 cells were treated with the phosphodiesterase (PDE) inhibitor IBMX (Figure 1G), the KCN-titration profiles under coupled and uncoupled respiratory conditions resembled what was observed in the untreated control cells (Figure 1H,I).

The specific inhibitory profile of an enzymatic step that is part of a measured metabolic flux can be exploited to assess the relative contribution of that step in controlling the flux [25]. On this basis, we thought to quantify the respiratory flux control coefficients of complex IV (C^J^_v,CIV_). To this aim, the reported inhibitory titration data sets were subjected to a non-linear regression analysis, as developed in [25,26] and modified in [18]. From the best fits, a C^J^_v,CIV_ of 0.08 ± 0.02 and 0.81 ± 0.08 was obtained for the OCR_Rest_ and OCR_Unc_, respectively, in the untreated HepG2 cells. The treatment using H89 resulted in a C^J^_v,CIV_ of 0.79 ± 0.08 and 0.75 ± 0.06 for the OCR_Rest_ and OCR_Unc_, respectively, whereas it was 0.15 ± 0.04 for the OCR_Rest_ and 0.85 ± 0.06 for the OCR_Unc_ in the IBMX-treated cells. The difference in the C^J^_v,CIV_ under the OCR_Rest_ between the untreated and H89-treated cells was statistically highly significant as well as between the OCR_Rest_ and OCR_Unc_ in the untreated and IBMX-treated cells, but not in the H89-treated cells.

To verify that the above-described treatment of the HepG2 cells using H89 effectively impacted on the cAMP–PKA axis, we carried out an immunoblot assay using an anti-P-Ser antibody on mitoplast-extracted proteins under native conditions in the presence of an amount of dodecyl maltoside that was reported to preserve the quaternary structure of all the RCs as well as their supramolecular aggregations [27]. Figure 1K shows a representative anti-P-ser immunoblot for the untreated and H89-treated HepG2 samples compared to the parental Coomassie blue stained gel. A clear stained anti-P-ser band can be observed in the untreated sample with an apparent molecular weight exceeding those of the RCs, as shown by the comparison with the Coomassie blue stained gel. Notably, in the H89-treated sample, this band was markedly dampened whereas it was present in the sample from the HepG2 cells treated using the adenylate cyclase-activating compound forskolin, with a staining intensity comparable to that of the untreated sample.

### 2.2. Inhibition of the Soluble Adenylate Cyclase Affects the Respiratory Flux Control Coefficients of CI, CIII and CIV under the Resting Respiratory State

Several isoforms of PKA have been described with a distinct intracellular compartmentalization [28]. Mitochondria-localized PKA have also been reported, but with the second messenger cAMP formed by a soluble HCO_3_^−^-dependent adelylate cyclase (sAC) [29,30]. Since sAC is specifically inhibitable by KH7 [31], we decided to replicate the inhibitory titration assay shown in Figure 1, but with HepG2 cells treated with KH7 (Figure 2A,B). Moreover, we extended our analysis to complex I and complex III using rotenone and myxothiazol as the specific inhibitors, respectively. The results of this systematic analysis are shown in Figure 2.

Figure 2C shows the KCN-related inhibitory profiles of the OCR_Res_ and OCR_Unc_ in the untreated control cells. The attained results confirmed what was already shown in Figure 1B, but were necessary because they were used as the internal control in a cohort of experiments carried out in a temporally distinct phase of this study. As for the results that were attained with the H89-treated cells as well as in the KH7-treated cells, the difference in the inhibitory profiles of the OCR under the resting and uncoupled conditions was practically abolished (Figure 2D) and confirmed in the OCR_Res_ vs. OCR_Unc_ plot (Figure 2E). When the OCR was titrated with the complex I inhibitor rotenone, the inhibitory titration profiles under the coupled and uncoupled conditions closely qualitatively resembled what was attained in the following KCN-titration in the untreated sample (Figure 2F). Most notably, treatment with KH7 resulted in similar inhibitory profiles irrespective of the respiratory state (Figure 2G,H). A behaviour similar to what was observed for the inhibitory titration curves with KCN and rotenone was recapitulated in the cells that were titrated with the complex III inhibitor myxothiazol (Figure 2I–K).

### 2.3. The Sum of the Respiratory Flux Control Coefficients of CI, CIII and CIV Indicates the Occurrence of Supercomplexes in Uncoupled Respiration and in the Presence of the Inhibitors of the cAMP/PKA Signaling Pathway

Applying the fitting procedure to the titration curves shown in Figure 2 enabled us to estimate the respiratory flux control coefficients of complexes IV (c^J^_v,CIV_), I (c^J^_v,CI_) and III (c^J^_v,CIII_). As shown in Figure 3A, in the untreated cells, the control coefficients estimated under resting respiration were significantly lower than those measured under uncoupled conditions for all the RCs (c^J^_v,CIV_: 0.30 ± 0.10 vs. 0.75 ± 0.07; c^J^_v,CI_: 0.19 ± 0.09 vs. 0.70 ± 0.15; and c^J^_v,CIII_: 0.39 ± 0.08 vs. 0.95 ± 0.04). Conversely, the treatment using KH7 resulted in relatively large control coefficients for all the RCs and, irrespectively, from the mitochondrial membrane energy state (c^J^_v,CIV_: 0.90 ± 0.05 vs. 0.90 ± 0.08; c^J^_v,CI_: 0.72 ± 0.17 vs. 0.75 ± 0.04; and c^J^_v,CIII_: 0.70 ± 0.12 vs. 0.70 ± 0.15). Most importantly, when the control coefficients of complexes I, III and IV were summed up in the different tested conditions, the sum was close to unity under resting respiration in the untreated cells. On the other hand, it largely exceeded unity (i.e., >2) in the uncoupled untreated cells as well as in the KH7-treated cells, irrespective of the respiration state (Figure 3B).

Figure 3C shows the effect of the drugs targeting the PKA-mediated signaling, which were tested in this study, on the uninhibited OCRs. While the maximal OCR_Unc_ in the untreated and IBMX-treated cells was, as expected, higher than the OCR_Rest_ in both the H89- and KH7-treated cells, the OCR_Unc_ was significantly inhibited when compared to the untreated cells. On the contrary, no significant differences were observed in the untreated and drug-treated cells as far as the OCR_Res_ was concerning. Consequently, the respiratory reserve capacity (i.e., the difference between the OCR_Unc_ and OCR_Res_), present in the untreated and IBMX-treated HepG2 cells, was practically zeroed in the H89- and KH7-treated samples.

### 2.4. Inhibition of the cAMP/PKA Axis Does Not Affect the Respiration-Driven Mitochondrial Membrane Potential Generation

To verify whether the attained results following the H89 or KH7 treatments were due to some compound-mediated uncoupling effect, we measured the mitochondrial membrane potential (ΔΨ_m_) under resting respiration. To this aim, the HepG2 cell membranes were permeabilized with digitonin (Figure 4A) supplemented with NADH-linked respiratory substrates and the ΔΨ_m_ was estimated from the fluorescence quenching of safranine. As shown in Figure 4B, both untreated and KH7-treated respiring HepG2 cells caused the quenching of the safranine-related fluorescence to a similar extent that was completely recovered following the addition of the uncoupler FCCP. From this outcome, the ΔΨ_m_ was quantified to be approx. 130 mV both in the untreated and KH7-treated cells. This result ruled out that the inhibitory titration profiles observed under resting (i.e., coupled) conditions in the H89 and KH7-treated cells, mimicking what was attained in the presence of FCCP, was trivially due to an uncoupling effect caused by the drug treatment.

### 2.5. A Number of PKA-Targeted Serine Residues Are Present at the Contact Sites of CI, CIII and CIV in the Supercomplex

Given the availability of several cryo-EM-resolved atomic structures in the respiratory chain SC, we decided to map the positions of the PKA-targetable residues. Figure 5A shows the porcine SC, which is currently the structure with the highest resolution (pdb 5GUP, 4.0 Å) [32]. The SC is constituted by one unit of complex I, interacting with a complex III dimer and a complex IV monomer (i.e., CI, CIII_2_, CIV). Exploiting the outcome of an extensive phospho-proteome bioinformatic analysis pinpointing, in all the RCs, all the known target residues for the serine/threonine/tyrosine protein kinases [20], we unbiasedly mapped the positions of only those that were targetable by PKA in the SC structure. Surprisingly, many of the PKA-targetable serine/threonine residues were located at the interfaces of the complexes, most notably at contact sites between CI–CIII, CIII–CIV and CIV–CI, and were potentially accessible from the matrix side of the membrane (Figure 5B).

Interestingly, some of these residues were proximal to salt bridges connecting CI and CIV (i.e., Ser14 in subunit 7C of CIV was close to the salt bridge formed between Arg20 in subunit 7C of CIV and Glu503 in subunit ND5 of CI) and to contact points linking CI to CIII (i.e., Ser26 of subunit NDUFB4/B15 of CI was close to Glu258, Glut259, Asp260 in Core I of CIII). Notably, the latter was recently reported to be essential for respirasome stabilization [33].

## 3. Discussion

The notion of a supramolecular organization of the mitochondrial RCs is supported by convincing evidence. Since the first reports identifying aggregations of RCs under native electrophoresis [4], the recent advance in cryo-EM analysis has contributed to the assessment of structural details of the SCs/respirasomes architecture at the atomic resolution [32,34]. Several aggregations of the complexes, with defined stoichiometries, have been identified. Those that contain one unit of complex I, a dimeric complex III and one unit of complex IV (complex II is not present) represent the more abundant assembly formulation [27]. Moreover, several biogenetic factors have been identified, facilitating the SCs maturation [35].

The acquisition of a supramolecular structure versus isolated units of the respiratory chain would confer additional functional properties, such as a facilitated electron transfer to the final acceptor O_2_. This would reduce the risk of electron diversion/leakage to O_2_ with the formation of potentially harmful reactive oxygen species [36]. Notably, all three complexes, I–III–IV, are redox-linked proton pumps, and the concerted locally generated transmembrane electrochemical potential could perhaps modulate its chemiosmotic coupling with the H^+^-F_o_F_1_-ATP synthase [37]. Complex II, which is not a redox-linked proton pump, is a free RC functionally interacting with SCs assemblies via a ubiquinone-mediated transfer of reducing equivalents.

Nevertheless, SCs coexist with a pool of isolated units comprising each of the RCs that constitute a reservoir from where the SC biogenesis taps. There is no reason to consider that this pool of isolated complexes is not competent in transferring reducing equivalents by a random collision mechanism. This infers the suggestive possibility that the isolated RCs are in a dynamic equilibrium with SCs as foreseen by a proposed “plastic” model of the respiratory chain [7,38]. Importantly, the prevailing metabolic state in the cell, as well as the bioenergetic state of the membrane, could shift the SCs vs. isolated RCs balance to better cope with the actual cellular energetic needs [10,17,18,39].

Proving the occurrence of the above-mentioned balance is experimentally challenging. Indeed, all the current methodological approaches for detecting the assembly state of the RCs rely on procedures that destroy the mitochondrial membrane’s integrity, therefore obscuring its functional state.

An alternative approach was provided by applying the metabolic flux theory, which assesses the specific contribution of the individual enzymatic steps in intact cells under different controlled functional states. In the context of the mitochondrial respiratory chain, the overall metabolic flux is given by the oxygen consumption rate fuelled by endogenous substrates. The easiest way to determine the individual contribution of the RCs is by measuring the impact of their specific inhibition on the respiratory flux [11,12,40]. The summation corollary of the theory foresees that the summation of the control coefficients of all the individual enzymatic steps, contributing to a given metabolic flux, cannot exceed the unity value unless two or more individual enzymatic steps assemble in supramolecular units.

In the framework of the metabolic flux analysis, our group showed in previous studies that the control coefficients of CI, CIII and CIV were markedly affected by the energy state of the mitochondrial membrane under conditions mimicking different metabolic states [17,18]. Applying the above-mentioned summation corollary, we proposed a model. In the presence of an established respiration-mediated transmembrane electrochemical potential, the RCs were prevalently in the form of isolated units, whereas under a low or absent membrane potential, the RCs assembled as respirasomes (see the model in Figure 6).

The recent improvement of the proteomic analysis for mitochondrial proteins disclosed a previously unappreciated occurrence of many different post-translational covalent modifications, including the proteins of the OxPhos system [19,20], confirming and extending the previously reported sporadic observations. These findings paved the way for understanding the impact of these modification on the structural/functional features of the proteins involved. The reversible phosphorylation of one or the other of the complexes of the OxPhos system is the better-documented post-translational modification [20,21,22,23].

The present study aimed to verify whether reversible covalent modifications had some effect in controlling the proposed balance between SCs and isolated RCs. We focused our attention on the phosphorylations mediated by the activation of the cAMP–PKA bio-signalling axis. It is worth noting that the standard protocol of cell culturing foresees the supplementation of a serum in the culture media to preserve cell viability. However, the large number of growth factors present in the serum maintain the constantly activated several signaling pathways, including the cAMP–PKA axis, which should be taken into account.

Herein, we tested the effect of a specific inhibitor of PKA on the mitochondrial respiratory activity of intact HepG2 cells and its impact on the inhibition titration profiles of complex IV using KCN as an inhibitor. Two conditions were tested (i.e., under a resting phosphorylating condition and under a fully uncoupled condition elicited by the protonophore FCCP). The attained results clearly showed that the inhibition of the PKA by H89 markedly affected the inhibition profile of complex IV on the overall resting respiratory flux, thereby abolishing the differences that were otherwise observed in the untreated cells under the two functional tested conditions. We ascribed this effect to the dephosphorylation of the respiratory chain complexes, which was likely caused by cellular protein phosphatases once PKA was inhibited. Interestingly, the treatment of cells with the phosphodiesterase inhibitor IBMX, which we expected to enhance the PKA activation, did not result in significant differences either in the respiratory flux or in the KCN-related inhibitory profiles, as compared to the untreated control cell. This confirmed that the composite mixture of the factors present in the serum-supplemented culture medium ensured a practically maximal upstream activation of the PKA-related signalling pathway [41,42].

Since several isoforms and intracellular compartmentalization forms of PKA, all inhibitable by H89, have been found, including intramitochondrial PKA [24], we further investigated its possible specific involvement in the findings reported herein. To this purpose, we exploited the notion that the activation of mitochondrial PKA is dependent on the production of cAMP by sAC, a distinct soluble isoform of the cytosolic transmembrane tmAC, which is selectively inhibited by KH7 [31]. Although sAC is not exclusively present in the mitochondria, it has been reported that, if the bioenergetic readouts are studied, the consequences following KH7 treatment relies largely on mitochondrial rather than the cytosolic sAC inhibition [43]. The treatment of the HepG2 cells with KH7 closely resembled what was observed with H89, regarding the dampening of the relative resistance to KCN in the inhibitory profile under resting respiration and its superimposition to that under uncoupled respiration. Most notably, a similar result was observed when the analysis was extended to both complex I and III. The quantification of the control coefficients of CI, CIII and CIV on the overall respiratory flux and application of the summation principle enabled us to conclude the following. (a) Under conditions favouring the phosphorylation state of the RCs, these functioned as isolated units under coupled conditions. As SCs, they functioned once respiration was uncoupled (a similar behaviour was observed following IBMX treatment). (b) Treatment with either the PKA inhibitor or the sAC inhibitor caused the RCs to function mainly as SCs under coupled conditions (see the schematic model presented in Figure 6). Our interpretation of the attained results agreed with a recent report, showing that the isoproterenol-mediated activation of the cAMP/PKA signaling cascade resulted in an enhanced formation of SCs in the H9c2 heart myoblast cell line [44].

The treatment using drugs inhibiting the PKA-mediated axis resulted, as reported, in a dampening of the maximal respiratory activity. Intriguingly, the respiratory activity under coupled phosphorylating conditions was apparently unaffected by either treatment, H89 or KH7. Therefore, it resulted in the complete abolishment of the respiratory reserve capacity. It was important to note that the KH7 treatment fully preserved the respiration-driven ΔΨ_m_ under coupled conditions. One possible interpretation of this puzzling observation was that, under resting conditions, the OxPhos system compensated for the defective respiratory activity, utilizing all the reserve available. The stimulated respiratory activity would then result in an independent ΔΨ_m_ and might be consistent with the proposed shift of the RCs from the isolated random collision state toward the more efficient SC state.

The biogenesis of the RCs (as well as their degradation) might influence the respirasome homeostasis. However, considering that the treatment with one or the other of the compounds affecting the cAMP/PKA signaling used in this study was relatively short (i.e., 2-h incubation), we would be inclined to consider the observed effects as unrelated to the RCs transcription/translation but rather to the post-translational reversible modifications.

A limitation of this study was that, although a much larger H89-sensitive immunoreactivity toward anti-P-serine was shown for high molecular weight SCs, we did not directly identify the phosphorylatable residues that were involved. Moreover, it cannot be ruled out that off-target effects might have been elicited, either linked to the specificity of the drugs used or to the phosphorylation/dephosphorylation of the proteins other than the RCs, thus resulting in indirect effects. This will be the object of future investigations. However, the exploration of the available cryo-EM structure of the CI, CIII_2_ and CIV respirasome to search for serine/threonine residues reported as a target of PKA [20] unveiled that most of them were mapped at the interlocked surfaces of the complexes and at contact sites that were largely exposed to the mitochondrial matrix side. Intriguingly, two phosphorylatable serines (i.e., Ser14 on sub. 7C of CIV and Ser26 on sub. NDUFB4/B15 of CI) were in close proximity to a salt bridge connecting CI to CIV or to a loop carrying three negative residues (i.e., Glu258-Glu259-Asp260) in the CIII core I subunit, which was recently reported to be essential in the stabilization of the respirasome CI/CIII_2_/CIV [39].

How the ΔΨ_m_ may influence the aggregation state of the RCs can, at the moment, only be the object of speculation, though computational MD modeling could be insightful. However, well-known membrane voltage-sensitive ion channels offer a biophysical prototype, demonstrating the impact of the electrical field nearby the membrane surface on the conformational change of the membrane-embedded proteins [45,46]. Considering that the electrical component of the proton motive force in the respiring mitochondria has been estimated to reach values of up to 180 mV at the inner mitochondrial membrane [2] (i.e., much larger than that measured at the plasma membrane of excitable cells), it is not surprising that it may exert strong electrostatic interactions with native or post-translationally added charges of protein residues, resulting in large conformational changes [47,48].

The cAMP/PKA-mediated signaling pathways have long been known to modulate many aspects of metabolism and energy balance, and once activated, they can promote glucose utilization, making available carbon sources for fuelling mitochondria respiration [49]. According to our model, the PKA-mediated phosphorylation of the RCs, modulating their dynamic equilibrium with respirasomes, made the respiratory activity sensitive to the bioenergetic state of the membrane, thereby improving the OxPhos efficiency. This “downstream” effect would cope with the “upstream” metabolic modulation elicited by the cAMP/PKA signaling cascade.

Together, the findings reported herein, if confirmed by other more direct approaches, might unveil a hitherto unappreciated level of plasticity and resilience in the mitochondrial OxPhos system.

## 4. Materials and Methods

### 4.1. Cell Culture

The human hepatoma-derived cell line (HepG2 from ATCC—HB-8065) was grown in DMEM (Dulbecco’s modified Eagle’s medium) and supplemented with 10% (*v*/*v*) foetal bovine serum to a 70–80% confluence before harvesting. Drugs treatments: 2 h using either 0.5 μM of *N*-[2-((*p*-Bromocinnamyl)amino)ethyl]-5-isoquinolinesulfonamide (H89), 25 μM of (E)-2-(1H-Benzo[d]imidazol-2-ylthio)-*N*′-(5-bromo-2-hydroxybenzylidene)propanehydrazide (KH7), or 100 μM of 3-Isobutyl-1-methylxanthine (IBMX), 10 μM of Forskolin (FK). After incubation, the cells were detached from the 150-mm diameter Petri dishes using 2 mL of 0.05% trypsin/0.02% EDTA and washed in 20 mL of phosphate buffer saline (PBS), pH 7.4, with 5% (*v*/*v*) calf serum, centrifuged at 500× *g*, re-suspended in 200 μL of PBS, counted and immediately used. The cell viability, as determined using the Trypan Blue exclusion, was typically never below 98%.

### 4.2. Respirometry

The oxygen consumption rate (OCR) was measured using high-resolution oxymetry (Oxygraph-2k, Oroboros Instruments, Innsbruck, Austria or Oxygraph+ System, Hansatech Instruments Ltd., Pentney, UK) with Clark-type oxygen electrodes in a thermostatically controlled chamber equipped with a magnetic stirring device and a gas-tight stopper fitted with a narrow port for additions via Hamilton micro-syringes. The calibration of the instruments was made according to the manufacturer instructions. The measurements were carried out at 37 °C with approx. 2 × 10^6^ HepG2 cells/mL suspended in 0.25 mM of sucrose, 10 mM of KH_2_PO_4_, 27 mM of KCl, 40 mM of Hepes, 1 mM of MgCl_2_, 0.5 mM of EGTA, 0.1% BSA and pH 7.1. The inhibitory titrations of the respiratory activities were performed under resting conditions (OCR_Rest_) or under uncoupled conditions in the presence of 0.5 mM f FCCP (OCR_Unc_) via sequential additions of 0.5 μL of freshly prepared and differently concentrated solutions of KCN (in ddH_2_O), myxothiazol (in ethanol) and rotenone (in ethanol). The addition of the vehicle ethanol at the highest final volume that was attained following titration did not result in any appreciable change in the OCR.

### 4.3. Metabolic Flux Control Analysis

The respiratory flux control coefficients (C^J^**_v,i_**) of the respiratory complexes I, III and IV were calculated using a non-linear regression analysis of the inhibitor titration data set, as developed in [25,26] and modified in [17,18] (fitting method). The derived non-linear equation that we utilized, correlating the percentage of the “global respiratory flux” to the inhibitor concentration [I], can be written as follows.
J/J_0_ (%) = 100 E/(c^J^_v,*i*_ (E_0_ − E) + E), with E = −0.5 (a − √(a^2^ + 4E_0_K_D_)) and a = [I] + K_D_ − E_0_

The above equation depended on three parameters: K_D_, which is the dissociation constant of the EI (enzyme inhibitor) complex; E_0_, which is the concentration of the active enzyme; and c^J^**_v_***_,i_*, which is the control coefficient of the inhibited enzymatic step (*i*). The K_D_s that were used were: 25 μM for the CIV–CN complex; 0.1 μM for the CIII–myxothiazol; and 0.4 μM for the CI–rotenone complex. The contents of the cytochrome c oxidase-aa_3_ (CIV) and bc_1_ complex (CIII) were evaluated using the dithionite-reduced minus air-oxidized differential spectra of the HepG2 cell lysate, resulting in 4.8 ± 0.5 pmol CIV/10^6^ cells (n = 5; Δε_650–630 nm_ = 24 mM^−1^ cm^−1^) and 2.1 ± 0.4 pmol CIII/10^6^ cells (n = 5; Δε_561–569 nm_ = 20 mM^−1^ cm^−1^). Thus, the values for E_0_ of 0.0025 μM and 0.001 μM were imputed for CIV and CIII, respectively, at the prevailing experimental conditions (i.e., 2 × 10^6^ HepG2 cells/mL). The amount of CI was assumed to be 0.8 pmol/10^6^ cells (E_0_ = 0.0004 μM), according to the relative respiratory chain complex ratios computed in [27]. The parameters of c^J^**_v_**_,I_ were estimated by finding the best fit of the inhibitory titration data set using the program GraFit 4.0.13 (Erithacus Software Ltd., West Sussex, UK), enabling a tolerance range of ±50% for the E_0_s and K_D_s values. The accuracy of the fitting method was further tested for the CIV complex, as described in [18].

### 4.4. Fluorimetric Measurement of ΔΨ_m_

A total of 3–5 × 10^6^ HepG2 cells in 2 mL of the respiration assay buffer (0.25 M sucrose, 10 mM KH_2_PO_4_, 27 mM KCl, 40 mM Hepes, 1 mM MgCl_2_, 0.5 mM EGTA, 0.1% BSA, pH 7.1) were assayed using oxymetry and treated with increasing concentrations of the plasma membrane permeabilizer digitonine (Dig) to assess the minimum amount of Dig inhibiting the resting respiratory activity. The amount of Dig chosen was for the fluorimetric analysis 20 μg/10^6^ cells. For the fluorimetric measurement (by FP-6500, Jasco Analytical Instruments, Easton, MD, USA), 1.5–2.0 × 10^6^ cells were pre-incubated for 20 min with the above reported concentration of Dig and pelleted at 1500–2000 rpm × 5 min. A total of 2 mL of the buffer was supplemented with 2 mM of pyruvate. In addition, 2 mM of malate was placed in the fluorimeter cuvette (λ_ex_ 495 nm, λ_em_ 596 nm, medium gain) and four consecutive additions of 2.5 μM of safranine O were injected to calibrate the fluorescence signal. After, 1.5–2.0 × 10^6^ cells (in 20 μL) were injected, causing progressive fluorescence quenching, followed by signal stabilization through the addition of 0.4 μM of FCCP, causing a full recovery of the safranine fluorescence. The estimation of the ΔΨ_m_ was performed, as described in [18]. Briefly, the following Nerst-derived equation was used.
ΔΨm(mV) = 60·log10[S]_in_/[S]_out_

where [S]_in_ stands for the intramitochondrial concentration of safranine, estimated from the fluorescence difference before and after the addition of FCCP [50], and [S]_out_ stands for the extramitochondrial concentration of safranine O, estimated from the fluorescence signal before the addition of FCCP. The value used for the intramitochondrial volume was 3 μL/10^6^ HepG2 cells, as computed in [18].

### 4.5. Blue Native-PAGE and Western Blotting

The mitochondrial-enriched fractions were obtained from the HepG2 cells using differential centrifugation. Briefly, 10^8^ cultured cells were harvested (by scraping) in 250 mM of sucrose, 1 mM of EDTA, 5 mM of Hepes and 3 mM of MgCl_2_ (pH 7.4), supplemented with 10 μL/mL of protease inhibitors stock solution and disrupted by tight Teflon glass homogenization. The homogenate was centrifuged at 1000× *g* for 10 min and the resulting supernatant at 14,000× *g* for 15 min. The resulting pellet was washed and finally re-suspended in a minimum volume of the same buffer supplemented with dodecyl maltoside (DDM) 0.6 g/g of protein. Blue native-PAGE was performed in the presence of Coomassie blue G250, as described in [27], in a 5–12% acrylamide gradient and a picture of the resulting gel was digitally acquired and stored. Hence, the gel was transferred onto a nitrocellulose membrane using an immersion electrophoretic transfer cell (Bio-Rad Laboratories S.r.l., Segrate, Italy) at 40 V overnight at 4 °C and blotted with mouse anti-phospho serine (P-ser) (Becton Dickinson, BD Headquarters Franklin Lakes, NJ, USA; 1:1000) and HPR-conjugated anti-mouse IgG (Thermo Fisher Scientific, Waltham, MA, USA; 1:20,000) as the primary and secondary Ab, respectively, according to the standard Western blotting procedure. Then, the gel was visualized using chemiluminescence (Chemidoc Imaging System, Bio-Rad ChemiDoc, Segrate, Italy).

### 4.6. Statistical Analysis

The data were reported as the mean (±the standard error mean, SEM) of at least three independent experiments, as indicated in the legends of the figures. The data were compared using an unpaired Student’s *t*-test or, when necessary, using two-way ANOVA followed by a post-hoc Bonferroni test. The differences were considered to be statistically significant when *p*-values < 0.05. All the analyses were performed using the GraphPad Prism Software Version 5 (GraphPad Software Inc., San Diego, CA, USA).

## Figures and Tables

**Figure 1 ijms-24-15144-f001:**
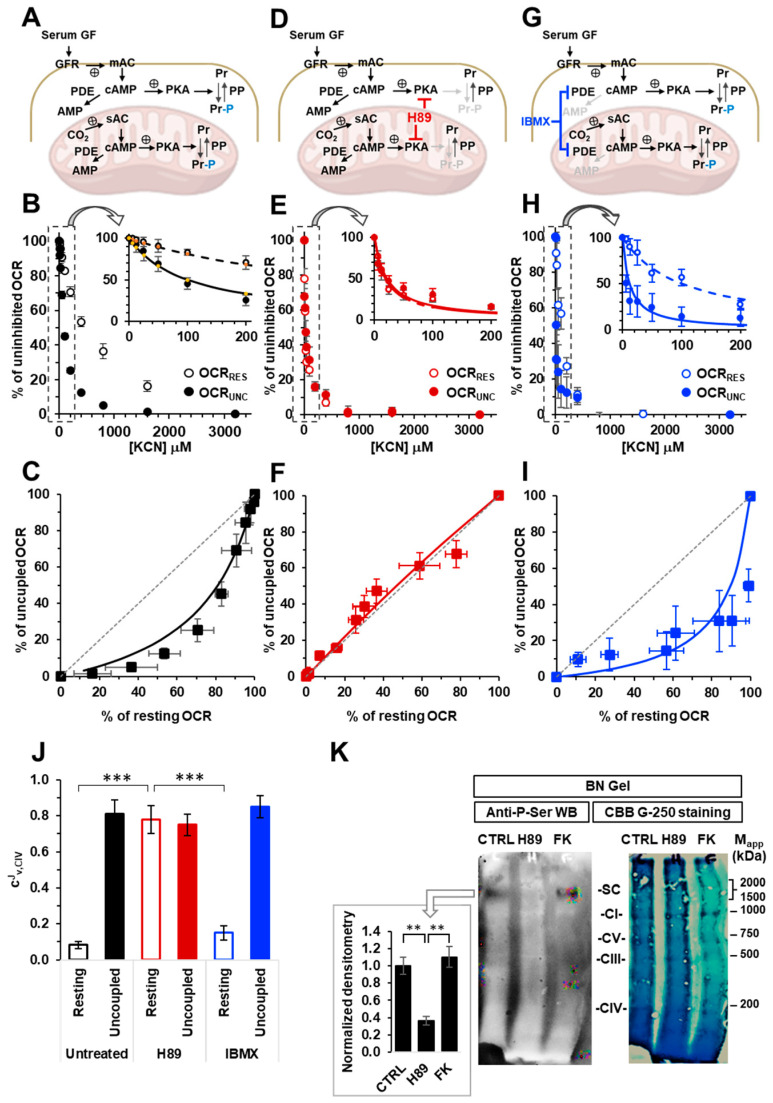
Effect of the modulators of the cAMP–PKA signalling axis on the KCN inhibitory titrations of the mitochondrial respiratory activity in the intact HepG2 cells. (**A**,**D**,**G**) schematically show the signalling pathway objects of the investigation under untreated conditions (**A**) and in the presence of H89 (**D**) or IBMX (**G**). GF, growth factors; GFR, growth factor receptor; mAC, membrane adelylate cyclase; PDE, phosphodiesterase; PKA, protein kinase A; PP, protein phosphatase; Pr, proteins; Pr-P, phosphorylated proteins; sAC, soluble adelylate cyclase. The endogenous substrates-sustained respiration in the HepG2 cells was assayed using high-resolution oxymetry in the absence or presence of 1 μM of FCCP as detailed in the Materials and Methods. The oxygen consumption rates were measured after the addition of approx. 2 × 10^6^ cells/mL and titrated with increasing concentrations of KCN. The oxygen consumption rates (OCRs) are shown as percentages of the uninhibited OCR at the concentrations of KCN tested in untreated (**B**,**C**), H89-treated (0.5 μM × 2 h) (**E**,**F**) and IBMX-treated (100 μM × 2 h) (**H**,**I**) cells. In （**B**,**E**,**H**）, the activity is shown under resting conditions (OCR_RES_ empty symbols) and under uncoupled conditions (OCR_UNC_, filled symbols) as the mean values ± the SEM of five to seven independent biological replicates under each condition. The insets detail the titration curves at relatively low and intermediate concentrations of KCN, with the dashed and continuous lines showing the best fit according to the equation described in the Materials and Methods. （**C**,**F**,**I** are the plots of the OCR_RES_ vs. OCR_UNC_, which were measured at the same concentrations of KCN with continuous lines resulting from the combination of those that fit the corresponding titration curves and the dotted thin line, indicating the diagonal of the diagram. (**J**) A histogram showing the flux control coefficient (c^J^_v,CIV_) of complex IV on the respiratory activity under resting and uncoupled conditions in untreated and H89- and IBMX-treated cells. The values are the means ± the SEM of the parameters computed from the best-fitting single titration curves averaged in (**B**,**E**,**H**); *** *p* < 0.001. (**K**) Representative immunoblotting with anti-P-serine for the mitochondrial-extracted proteins separated by BN-native gel electrophoresis (shown on the right as stained by Coomassie blue brilliant (CBB)-G250) as detailed in the Materials and Methods. The histogram shows the densitometric analysis of the immunostained high molecular weight band as the mean ± the SEM of three independent biological replicates under each condition; ** *p* < 0.005 (CTRL, untreated cells; H89-treated cells; FK, forskolin-treated cells).

**Figure 2 ijms-24-15144-f002:**
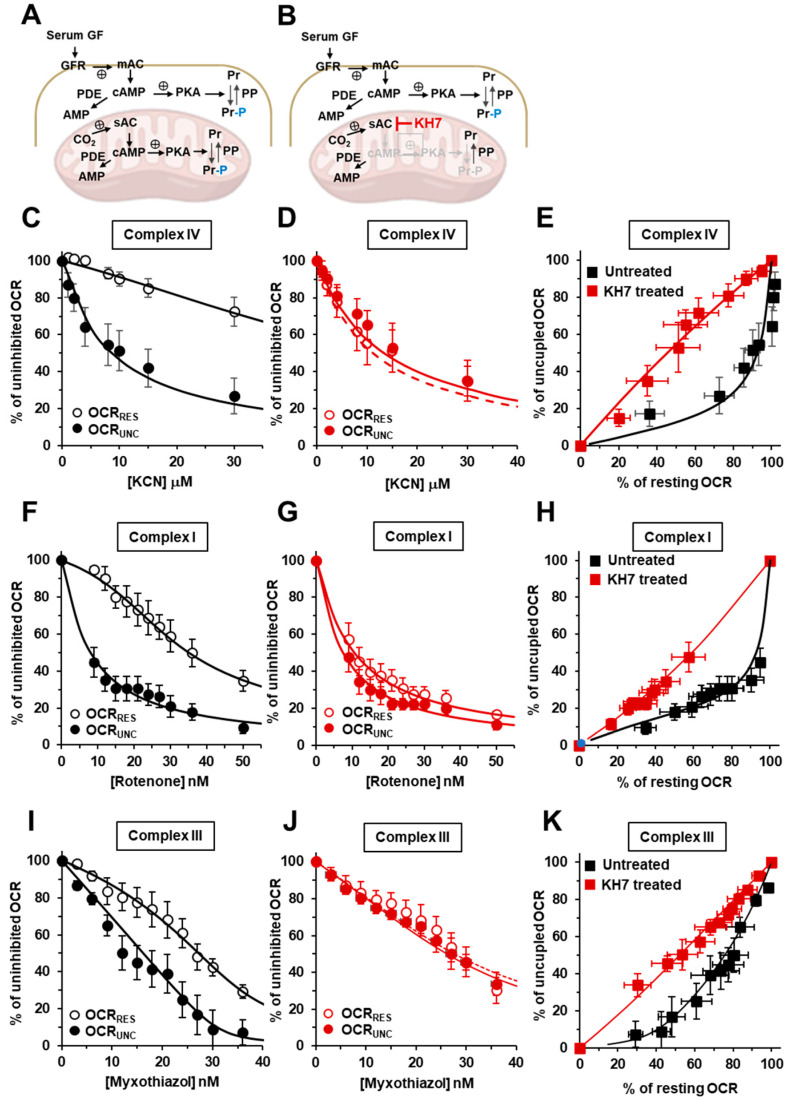
Effect of the inhibition of sAC by KH7 on the KCN, rotenone and myxothiazol inhibitory titrations of the mitochondrial respiratory activity in the intact HepG2 cells. (**A**,**B**) schematically show the signalling pathway object of the investigation under untreated conditions (**A**) and in the presence of KH7 (**B**). See the legend in Figure 1 for the definitions of the abbreviations. The endogenous substrates-sustained respiration in the HepG2 cells was assayed using high-resolution oxymetry in the absence or presence of 1 μM of FCCP as detailed in the Materials and Methods. The oxygen consumption rates (OCRs) were measured after the addition of approx. 2 × 10^6^ cells/mL and titrated with increasing concentrations of either KCN, rotenone or myxothiazol. The OCRs are shown as percentages of the uninhibited OCR at the relatively low and intermediate concentrations of KCN (**C**), rotenone (**F**) and myxoyhiazol (**I**) in untreated cells and in KH7-treated cells (25 μM × 2 h) (**D**,**G**,**J**) respectively. The activity is shown under resting conditions (OCR_Res_ empty symbols) and under uncoupled conditions (OCR_Unc_, filled symbols) as the mean values ± the SEM of five to seven independent biological replicates under each condition with the dashed and continuous lines showing the best fit, according to the equation described in the Materials and Methods. (**E**,**H**,**K**) are the plots of the OCR_RES_ vs. OCR_Unc_ measured at the same concentrations of KCN, rotenone and myxothiazol, respectively, with continuous lines resulting from the combination of those that fit the corresponding titration curves.

**Figure 3 ijms-24-15144-f003:**
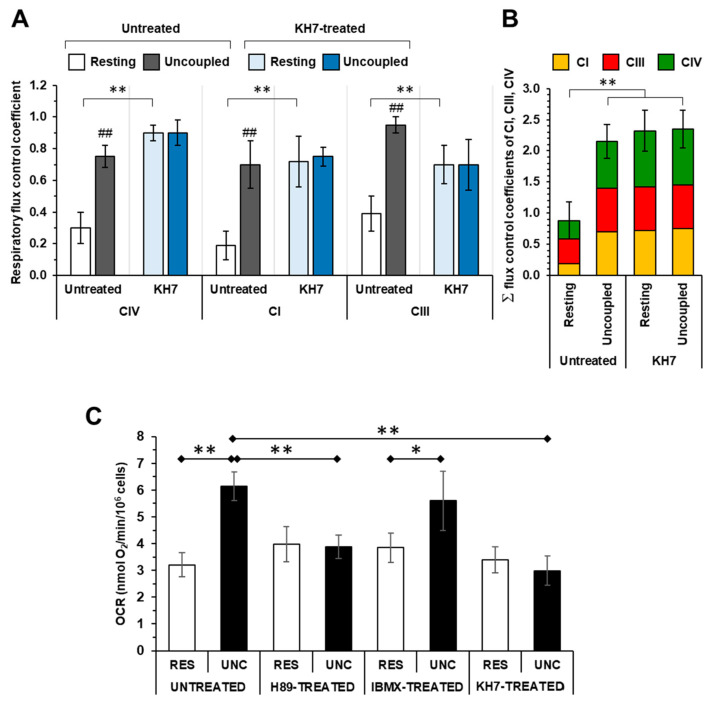
Effect of the sAC inhibition on the respiratory flux control coefficients of CI, CIII, CIV and of the cAMP–PKA modulation on the uninhibited oxygen consumption rate in the HepG2 cells. (**A**) The respiratory flux control coefficients of CIV, CI and CIII under resting and uncoupled conditions, as computed using the inhibitory titration curves shown in Figure 2C,F,I for the untreated cells and in Figure 2D,G,J for the KH7-treated cells. The values are the means ± the SEM of the parameters computed from the best-fitting single titration curves (n = 5–7) averaged in Figure 2; ** *p* < 0.01; ^##^
*p* < 0.01 compared with resting. (**B**) Histogram showing the sum of the control coefficients of CI, CIII and CIV under uncoupled and coupled respiration in the untreated and KH7-treated cells. (**C**) Histogram showing the effect of the H89, IBMX and KH7 treatment on the oxygen consumption rates (OCR) in the cells under resting (RES) and uncoupled (UNC) respiratory conditions. The OCR was measured polarographically as detailed in the Materials and Methods and the bars are the means ± the SEM of seven to ten independent biological replicates under each condition; * *p* < 0.05; ** *p* < 0.01.

**Figure 4 ijms-24-15144-f004:**
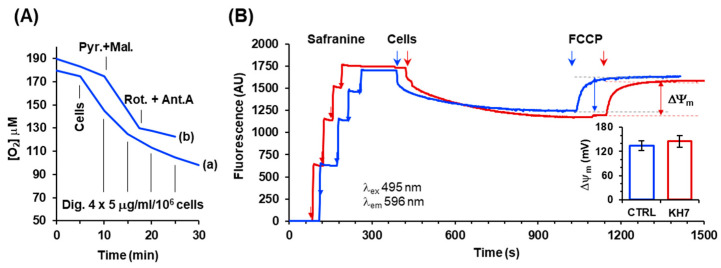
Measurement of the respiration-driven mitochondrial membrane potential in the permeabilized HepG2 cells. (**A**) Representative oxymetric traces showing the protocol utilized to permeabilize the plasma membrane in the HepG2 cells. Trace (**a**): 2 × 10^6^ cells/mL were added to the assay buffer (detailed under the Material and Methods) and supplemented as indicated with consecutive additions of digitonin (Dig). Trace (**b**): 2 × 10^6^ cells/mL were pre-incubated with 15 mg/mL/10^6^ cells and digitonin. They were suspended in the assay buffer and supplemented, where indicated, with 2 mM of pyruvate plus 2 mM of malate followed by an injection of 2 mM of rotenone plus 1 mM of antimycin A. (**B**) Representative spectrophotometric traces for the assessment of the mitochondrial membrane potential (ΔΨ_m_) in the untreated (blue line) and KH7-treated (red line) cells. Where indicated, the assay buffer supplemented with pyruvate and malate was injected with four consecutive additions of 2.5 mM of safranine O followed by addition of the previously digitonin-permeabilized cells (2 × 10^6^/mL). After the stabilization of the quenched fluorescence, 0.5 mM of FCCP was added. The fluorescence difference recorded before and after the addition of FCCP is a measure of the respiration-driven ΔΨ_m_ generation. The histogram in the inset shows the quantification in mV of the ΔΨm, as described in the Materials and Methods, with the bars representing the means ± the SEM of the four independent biological replicates.

**Figure 5 ijms-24-15144-f005:**
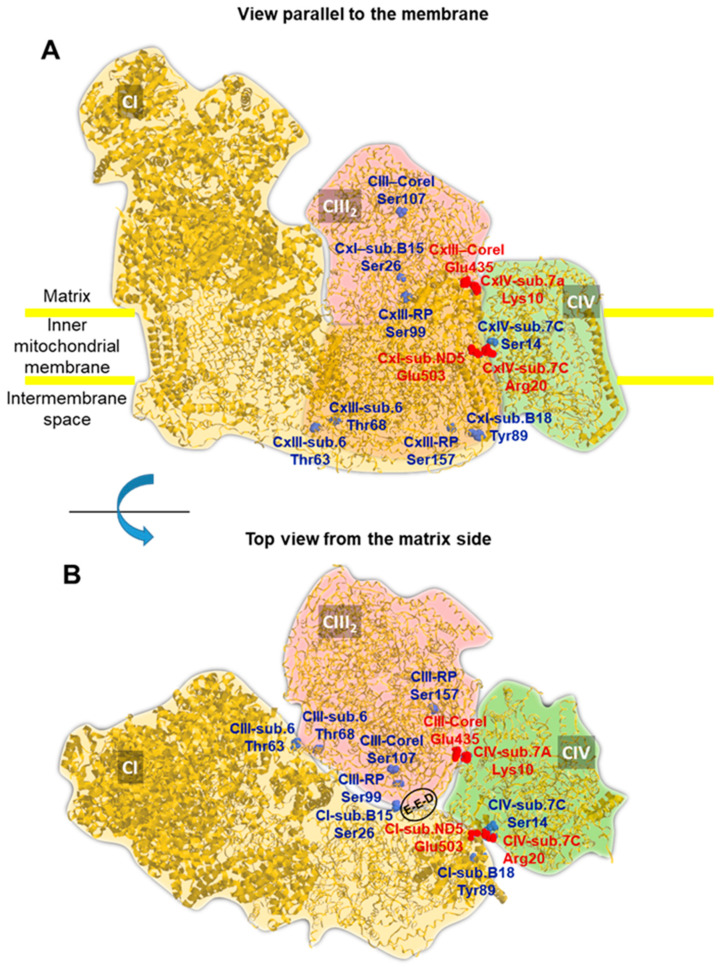
Structure of the mitochondrial respirasome with the localization of the PKA-targetable residues. The cryo-EM-derived structure of the mitochondrial respiratory SC CI/CIII_2_/CIV from *Sus scrofa* is shown (pdb code 5GUP). The silhouettes of the individual complexes are highlighted using different colors, and the view parallel to the membrane (**A**) and the top view from the matrix side (**B**) are shown. The serine and threonine residues targeted by PKA are pinpointed in the structure (in blue) with an indication of the complex subunit of their location. The numbering of the residues corresponds to the primary sequence of the specific subunit in *Homo sapiens* after an alignment to that of the porcine sequence. Two inter-complex salt bridges between complexes I–IV and III–IV are also shown in red. The encircled residues E-E-D indicate Glu258-Glu259-Asp260 in the Core I of CIII. See the text for further details. The picture was drawn using RasTop 2.2 (geneinfinity.org/rastop/ (accessed on 7 July 2023)).

**Figure 6 ijms-24-15144-f006:**
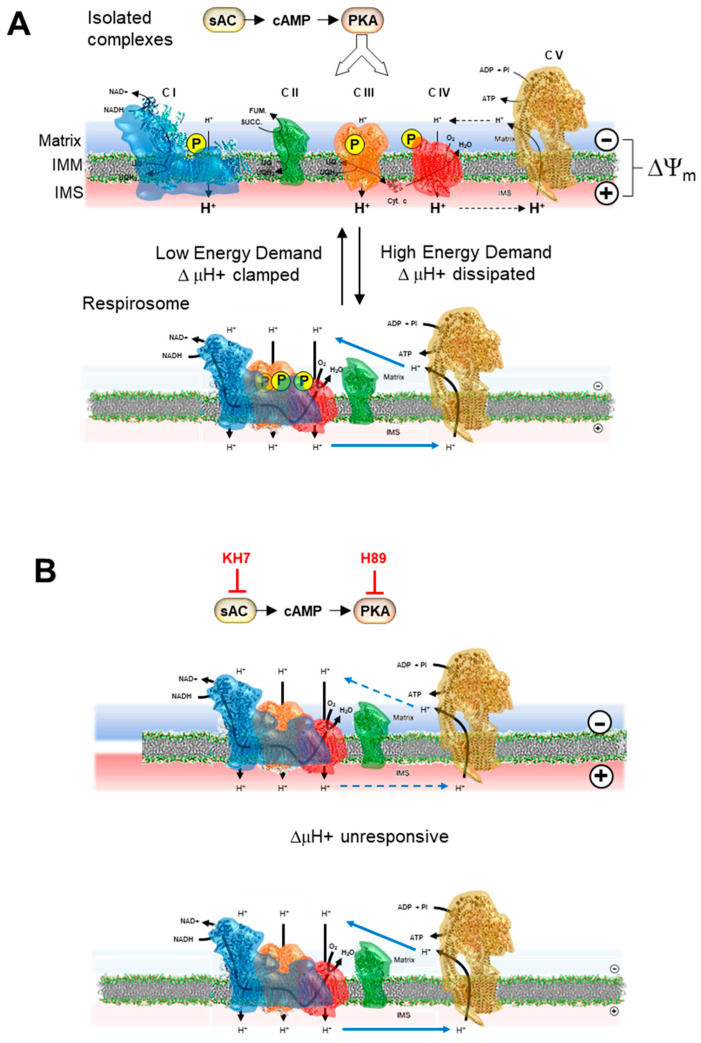
PKA-mediated control of the respiratory chain complexes/supercomplex equilibrium in the mitochondrial membrane. Pictorial representation of the model proposed in this study. The respiratory chain complexes are shown in their phosphorylated state under the action of the mitochondrial sAC/cAMP/PKA signaling axis (**A**). These complexes would change their aggregation state as a function of the bioenergetic state of the membrane/metabolic demand, with the isolated complexes state favored by the presence of a large electrochemical membrane potential (ΔμH^+^ largely contributed by ΔΨ_m_). The formation of supercomplexes is elicited by low ΔμH^+^. (**B**) highlights that the inactivation of the signaling pathway and the consequent dephosphorylation of the complexes causes their tendency to form supercomplexes irrespective of the ΔμH^+^ extent. IMM, inner mitochondrial membrane; IMS, intermembrane space. See the Discussion for further details.

## Data Availability

All the research data are provided in the publication.

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
