# Peer review of "Mitochondrial sAC-cAMP-PKA Axis Modulates the ΔΨ_m_-Dependent Control Coefficients of the Respiratory Chain Complexes: Evidence of Respirasome Plasticity"

_ijms, 2023, doi:10.3390/ijms242015144_

Round 1

Reviewer 1 Report

The manuscript presents and discusses the obtained results in a clear, sound way. I have only several remarks:

1. Image 1K - electrophoresis images of better quality shall be presented. Did the authors try to use the BN approach to check the supercomplex organisation upon the uncoupler treatment? It maybe tricky as some of the immediate effects of the uncoupler may be not seen during sample preparation procedure, but maybe some effects could be seen. Verification of the hypothesis with different technique would largely increase the strenght of the presented evidence.

2. The schemes presenting sites of action of the used inhibitors help to follow the experimental design. They shall be made bigger to improve their readability.

3. line 118 - wrong figure numbers seem to be mentioned ("Fig 3A, B")

4. Figure 4 - it is not clear how the experiments refers to the investigated topic, it is not described in the text.

Author Response

Comments and Suggestions for Authors - Reviewer #1

The manuscript presents and discusses the obtained results in a clear, sound way. I have only several remarks:

  1. Image 1K - electrophoresis images of better quality shall be presented. Did the authors try to use the BN approach to check the supercomplex organisation upon the uncoupler treatment? It maybe tricky as some of the immediate effects of the uncoupler may be not seen during sample preparation procedure, but maybe some effects could be seen. Verification of the hypothesis with different technique would largely increase the strenght of the presented evidence.

As required, we improved the quality of the image in Fig. 1K enhancing homogenously the contrast. We tried several times to compare the BN gel profiles of proteins extracted from mitochondria pre-incubated with respiratory substrates in the absence and in the presence of one or the other of uncoupler (FCCP), ATP-synthase inhibitor (oligomycin), PKA inhibitor. In all the conditions tested no apparent difference was observed. As commented in the Discussion (lines 368-371 new manuscript) we are convinced that the procedural steps following pre-treatment of cells as well as of isolated mitochondria destroy the membrane integrity and rapidly shift the pre-existing dynamic equilibrium between RCs and SCs toward a condition mimicking the fully uncoupled functional state.   

Perhaps FRET-related methodologies would be an alternative more conservative approach. In this case genetically engineered RCs should be generated, isolated, and incorporated in liposomes following their binding to suited fluorophores to assess their interaction tendency under different conditions mimicking physiological cues. These quite elaborate experiments are within our research perspectives, but they need time.

  1. The schemes presenting sites of action of the used inhibitors help to follow the experimental design. They shall be made bigger to improve their readability.

As required the schemes shown in Figs. 1 and 2 have been accordingly enlarged as well as the entire Figures.

  1. line 118 - wrong figure numbers seem to be mentioned ("Fig 3A, B")

The reference to Figs. 3A, B made in the Results section at line 118 is not a wrong citation but an anticipation of the Figures whose content will be more amply discussed later.

  1. Figure 4 - it is not clear how the experiments refers to the investigated topic, it is not described in the text.

We thank the reviewer for this observation. In the introduction to the section 2.4. we explained the rationale of the experiment, but perhaps we were not clear enough. Therefore, we modified the text highlighting that the measurement of the mitochondrial membrane potential was needed to verify if treatment with the inhibitor of the sAC/cAMP/PKA cascade had some uncoupling effect on respiration (lines 281-283 in red in the new manuscript). The results attained ruled out this possibility.

Reviewer 2 Report

Authors analyzed the potential role of different respirasome complexes in hepatic cells. The respirasome formation may be of high interest and could be different in other cells depending on substrate availability and energetic requirement, among other factors. Thus, the work suggests an interesting area of research. However, some questions should be further discussed.

-       firstly, the specificity of used inhibitors (IMBX, H89) may lead to indirect effects unrelated with proposed targets. By silencing these genes, authors could have confirmed the proposed pathways. Please, include as limitation.

-       How would be the influence of respirasomes on complex II and ATPase? Would be respiration deviated to lactate synthesis? Please, discuss.

-       Respirasomes could be influenced by mitochondrial transcription factors and coactivators such as PPARs, PGC1a, NRF, TFAM-1. Similarly, the expression of mitochondrial genes may be altered. Please discuss.

-       The size of figures 5 and 6 should be bigger

Author Response

Comments and Suggestions for Authors – Reviewer #2

Authors analyzed the potential role of different respirasome complexes in hepatic cells. The respirasome formation may be of high interest and could be different in other cells depending on substrate availability and energetic requirement, among other factors. Thus, the work suggests an interesting area of research. However, some questions should be further discussed.

-       firstly, the specificity of used inhibitors (IMBX, H89) may lead to indirect effects unrelated with proposed targets. By silencing these genes, authors could have confirmed the proposed pathways. Please, include as limitation.

We agree with the reviewer’s observation that off-target effects might have been elicited either linked to the specificity of the drugs used or to phosphorylation/de-phosphorylation of proteins other than the RCs resulting therefore in an indirect effect. This point has now been commented on in the revised version (lines 475-478, in red).

-       How would be the influence of respirasomes on complex II and ATPase? Would be respiration deviated to lactate synthesis? Please, discuss.

We thank the reviewer for these stimulating observations that we did not consider in our paper because out of its specific scope. However, as far as complex II is concerned, no evidence supports its participation to respirasome/SCs formation. Complex II would be a free complex functionally interacting with SCs via ubiquinone-mediated transfer of reducing equivalents (commented in lines 358-360). The same notion applies to the ATP-synthase complex (CV) that is not part of SCs. In this latter case as mentioned in the Discussion (lines 357-359 in red) concerted “localized” protonmotive force elicited by the respirasome proton pumping complexes might have some effect on the efficiency of FoF1-ATP synthase. However, this point, at the moment, is just speculative. Interestingly, phosphoproteomic analysis as well as dedicated investigations have identified several PKA-related phosphorylation sites also in CV (see Ref. 20 in the main text).

As pertinently alluded by the reviewer, metabolism is long known to be modulated by the cAMP/PKA signaling pathway that once activated promotes, among other, glucose utilization making available carbon source for fueling mitochondria respiration.  According to our model the PKA-mediated phosphorylation of the RCs, modulating their dynamic equilibrium with respirasomes, makes the respiratory activity sensitive to the bioenergetic state of the membrane thereby improving the OxPhos efficiency. This “down-stream” effect would cope with the “up-stream” metabolic modulation elicited by the cAMP/PKA signaling cascade. This point is commented on in the discussion of the revised manuscript (lines 497-504 in red) and a new reference added (ref. 50).

-       Respirasomes could be influenced by mitochondrial transcription factors and coactivators such as PPARs, PGC1a, NRF, TFAM-1. Similarly, the expression of mitochondrial genes may be altered. Please discuss.

We agree with this point highlighted by the reviewer, that biogenesis of the RCs (as well as their degradation) might influence the respirasome homeostasis. However, considering that the treatment with one or the other of the compounds affecting the cAMP/PKA signaling, used in this study, was relatively short (i.e. 2 hours incubation) we would be inclined to consider the observed effects as unrelated to RCs transcription/translation but rather to post-translational reversible modifications. This point has been commented on in the revised version (lines 468-472, in red).

-       The size of figures 5 and 6 should be bigger.

As required both Figs. 5 and 6 have been accordingly enlarged as well as the font size of the indicated residues in Fig. 5.